# Disparities in childhood composite index of anthropometric failure prevalence and determinants across Ethiopian administrative zones

**Haile Mekonnen Fenta** [1]*, **Temesgen Zewotir**[2], **Essey Kebede Muluneh**[3]

**1** Department of Statistics, College of Science, Bahir Dar University, Bahir Dar, Ethiopia, **2** School of Mathematics, Statistics and Computer Science, College of Agriculture Engineering and Science, University of KwaZulu-Natal, Durban, South Africa, **3** School of Public Health, Bahir Dar University, Bahir Dar, Ethiopia

* hailemekonnen@gmail.com

## Abstract

### Background

The prevalence of under-five children's undernutrition in Ethiopia is among the highest in the world. This study aimed at exploring the prevalence and risk factors of the composite index for anthropometric failure (CIAF) of under-five children in Ethiopia by incorporating the zonal (district) effects.

### Methods

The data was drawn from Ethiopian Demographic and Health Surveys (EDHSs), a population-based cross-sectional study of 29,599 under-five year children from 72 Zones in the years 2000, 2005, 2011, and 2016. Fixed effect variables related to child and maternal-household were included in the model. We adopted a generalized mixed model with CIAF as outcome variable and Zones as random effects.

### Results

The prevalence of CIAF in Ethiopia was 53.78% with the highest prevalence of 61.30% in 2000 and the lowest prevalence of 46.58% in 2016. The model result revealed that being a female child, absence of comorbidity, singleton births, and the first order of birth showed significantly lower CIAF prevalence than their counterparts. Among the household characteristics, children from mothers of underweight body mass index, uneducated parents, poor household sanitation, and rural residents were more likely to be undernourished than their counterparts. Based on the best linear unbiased prediction for the zonal-level random effect, significant variations of CIAF among zones were observed.

### Conclusion

The generalized linear mixed-effects model results identified gender of the child, size of child at birth, dietary diversity, birth type, place of residence, age of the child, parental level

**Data Availability Statement:** The data used in this study are third party data from the DHS program (https://dhsprogram.com/data/available-datasets.cfm) and can be accessed following the protocol

outlined in the Methods section. The shapefile of the map of Ethiopia was accessed as an open-source without restriction from open Africa 2016 (https://africaopendata.org/dataset/ethiopia-shapefiles).

**Funding:** The authors received no specific funding for this work.

**Competing interests:** The authors have declared that no competing interests exist.

**Abbreviations: BMI**, Body Mass Index; **CI**, confidence interval; **CIAF**, Composite Index for Anthropometric Failure; **CSA**, Central Statistics Authority; **EDHS**, Ethiopian Demographic and Health Survey; **GIS**, Geographic Information System; ICC, Intra-class correlation; **SNNP**, Sothern Nation and Nationalities and People; **SSA**, Sub-Saharan Africa; **U5C**, under five-children.

of education, wealth index, sanitation facilities, and media exposure as main drivers of CIAF. Disparities of CIAF were observed between and within the Ethiopian administrative Zones over time.

## Introduction

Childhood malnutrition, which occurs in the form of undernutrition and overnutrition affects the economic, social, and medical well-being of individuals and households [1–3]. In low-middle-income countries including Ethiopia, undernutrition is the most common form of malnutrition and a leading cause of death in children [1–8]. The survivors of undernourished children suffer from mental and physical problems and ultimately affect the overall economy.

Commonly, stunting, wasting, and under-weight have been conventionally utilized to assess the prevalence of under-nutrition among children. The current body of evidence of the prevalence and determinant factors of undernutrition in Ethiopia has focused on a single undernutrition measure such as wasting, stunting, and being underweight [4–9]. However, those conventional indices used alone grossly underestimate prevalence mainly due to the overlapping of the children into multiple categories of anthropometric failure failed to give true estimates of the real burden of childhood undernutrition. This is because; the commonly used indices may overlap that the same child could show signs of having two or more of these undernutrition indicators simultaneously, and hence insufficient for determining the overall real burden of undernutrition situations among under-five children [5–7, 9–16]. Countries such as China, India, Malawi, Bangladesh, and others were adopted the CIAF model as their children's undernutrition status [6, 9–13].

The CIAF is computed by grouping children whose height and weight are above the age-specific norm (above -2 z-scores) and also children whose height and weight for their age below the norm and thus experiencing one or more forms of anthropometric failure as such: B-wasting only, C-wasting and underweight, D- wasting, stunting and underweight, E- stunting and underweight, F-stunting only and Y- underweight only. The CIAF is then calculated by aggregating these six (B-Y) categories [5, 7, 14–16].

The cause for undernutrition in under-five children is complicated and multifaceted. To date, numerous studies have been done to examine the determinants of undernutrition in different countries. Those factors include food insecurity, poor socio-economic conditions, socio-demographic characteristics, poverty, presence of comorbidities, inadequate feeding practices, sanitation problems, breastfeeding, and inadequate complementary foods [4–16]. Though a number of studies demonstrated that Ethiopia has recorded promising progress in reducing levels of under-nutrition over the past two decades, the challenges and achievements of different administrative Zones were not studied yet. Detecting the problem of under-nutrition and its variation among administrative Zones provides deeper insight into the countries health priorities for under-five children and for zonal health departments to plan, follow up, monitor, and evaluation of health activities at the lower level. As there are some cultural and climatic variations among administrative Zones, which result in different practices regarding staple food in the Zones, it would be important to assess the CIAF at the Zone level [17–20].

The objective of this study was, therefore, to explore the prevalence and identify risk factors of childhood CIAF by accounting for the differences in the effects of administrative Zones using the data from 2000, 2005, 2011, and 2016 EDHSs.

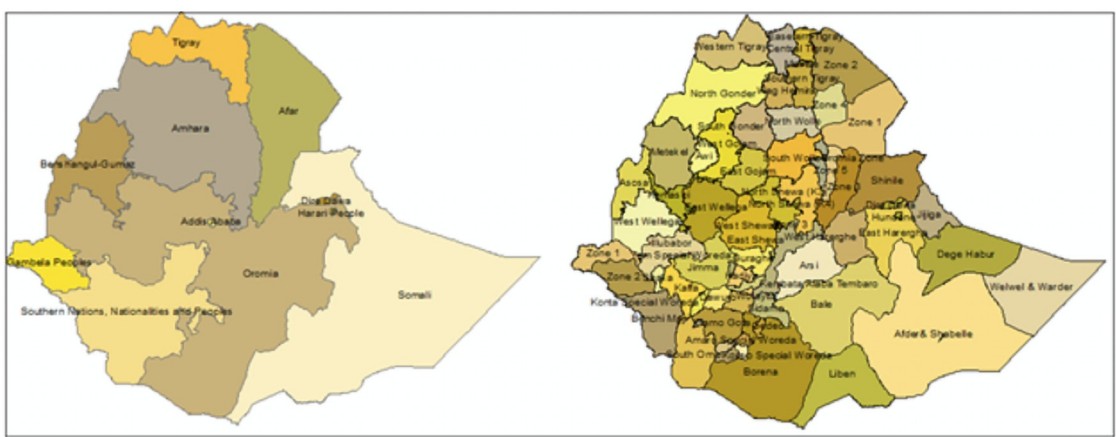

**Fig 1. Map of Ethiopia and showing 11 regions and 72 zones.**

## Materials and methods

### Settings

The data for this paper was drawn from the nationally representative cross-sectional study design of 2000, 2005, 2011, and 2016 EDHS. Ethiopia, which is a low-income country in East Africa, is a completely landlocked country that has a surface area of 1.1 million km$^2$ [21]. For administrative purposes, the country is divided into 11 regions; and a total of 72 administrative areas called zones, a setting for which the entire analysis is carried out (Fig 1).

### Data sources

The study used the birth history record of EDHS collected in 2000, 2005, 2011, and 2016. This study was conducted on 29,599 children consisting of 8,139 from 2016, 9,238 from 2011, 3,767 from 2005, and 8,455 children from the 2000 EDHS respectively. For the sampling frame classification, both the 1994 and 2007 population and housing census were used [22, 23].

### Variables

The outcome variable of this study was the composite index for anthropometric failure [5, 7, 14–16]. The risk factors comprise child, women, and household-related levels selected based on findings in the literature [8, 9, 24–29]. The dietary diversity score (DDS) of children was computed from seven food groups based on the EDHS data. These included (grains, roots, and tubers; legumes and nuts; dairy products (cheese, milk, and yogurt); flesh foods (meat, fish, poultry); eggs; vitamin A-rich fruits and vegetables) and other fruits and vegetables) [30–33]. If a child consumed at least one food item from each of the food groups, we assigned "1" else "0". The acceptable minimum DDS has involved diets from at least four food groups [30–33]. The other important variable is women's autonomy which was measured from three indicators: women's attitude towards wife punishment (5 measures), property ownership (2 items), and involvement in decision-making (4 items). Finally, the principal component analysis (PCA) method was used to generate autonomy scores. After the score has been constructed, the autonomy variable was then recorded as tertiles with categories labeled low, middle, and high autonomy [34–38].

Moreover, though the wealth index variable for the 2000 EDHS was not available and it was calculated from the data. The study participants of the EDHS dataset are from both rural and

urban areas, the source of assets varies from rural to urban dwellers, and then we take into account this difference to compute the wealth index. Items including (water, toilet, electricity, radio, television, telephone, refrigerator, motorcycle/scooter, car/truck, cattle, floor, own house, cropland) were used to measure the WI. Most of the items are common for both rural and urban. But some items like "owing cattle" seems more common in rural and some item like the source of energy (electricity) common in urban. Principal component analysis (PCA) has been conducted to generate the WI cores for urban and rural separately and then merging (mixing) them together. We divide the factor scores into five equal parts (quintiles) [39, 40].

## Statistical methods

Supposing our data is clustered and the response variable denoted by $y_{ij}$, where i = 1,2,3,. . ., 29,599 children and j = 1,2,3,. . . 72 zones. Depending on a vector of individual random effects $u_i$, the outcome variables are assumed to be independent, with density functions belonging to the exponential family [41–44].

$$f\left(y_{ij}|\theta_{ij}, \phi\right) = \exp\left[\phi^{-1}\left\{y_{ij}\theta_{ij} - \psi\left(\theta_{ij}\right)\right\} + c\left(y_{ij}, \varphi\right)\right],$$

where $\varphi$ is a scale parameter, $c(.)$ is a function only depending on $y_{ij}$ and $\varphi$, and $\psi(.)$ is a function satisfying $E\left(y_{ij}|u_i\right) = \psi'\left(\theta_{ij}\right) = v(x_{ij}^T\beta + z_{ij}^T u_i)$ and $\text{Var}(y_{ij}|u_i) = \phi\psi''(\theta_{ij})$, for which $v(.)$ denotes a known link function, $x_{ij}$ and $z_{ij}$ are vectors of covariates, $\beta$ is a vector of unknown fixed effect parameters.

Hence, in this study, we adopted the generalized linear mixed model (GLMM) [45–49] to examine the effect of the child, woman, and household characteristics on CIAF undernutrition measures for under-five children in Ethiopia. The adopted GLMM model is:

$$g\left(\mu_{ij}\right) = logit\left(\mu_{ij}\right) = \log\left(\frac{\mu_{ij}}{1 - \mu_{ij}}\right) = \log\left(\frac{P\left(y_{ij} = 1\right)}{P\left(y_{ij} = 0\right)}\right) = \eta_{ij},$$

an alternative link function is probit link $\phi^{-1}(\mu_{ij})$, the inverse standard normal cumulative distribution function.

Where $\eta_{ij} = \beta_0 + \beta_1 x_{1ij} + \cdots + \beta_k x_{kij} + u_{0j}$, $X_{1ij}$, through $X_{kij}$ denote the k explanatory variables measured on children, women, and households. $\mu_{ij}$ and $1-\mu_{ij}$ are respectively the probability of a child getting CIAF and not having CIAF (j = 1,. . ., 72 Zones, i = 1,. . ., $n_j$ children within each Zone).

$\beta_0$ is the log odds of intercept

$\beta_1 \ldots \beta_k$ are effect sizes of children and household-level covariates

$u_{0j}$ are random errors at Zone level

The distribution of $u_{0j} \sim N(0, \sigma_{u0}^2)$. The intra-class correlation (ICC) was computed using between-Zone variance and the within the Zone, variance (ICC = $\left(\sigma^{u2}/_\sigma u^2 + \sigma_e^2\right)$ [42–44].

The mixed model approach permits the estimation of the fixed effects using the Best Linear Unbiased Estimation (BLUE), and the prediction of the random effects using the Best Linear Unbiased Prediction (BLUP) procedure by solving a generalized form of mixed equations [46, 50–52]. The BLUP is a realized value of the random effects [53, 54], which provides an unbiased method by adjusting the known sources of variation [51]. BLUP is a commonly used method to predict genetic and breeding values in both animals and plants. It is used to rank and select the best plants [55–58], for selecting and ranking animals for breeding [59, 60]. It is also used to rank the performance of employees in a certain organization [54]. In our study,

**Table 1. Classification of undernourished under-five children and their percentages over time in Ethiopia.**

| Group | Description of the group | Definition | Wasting | Stunting | Underweight | 2000 | 2005 | 2011 | 2016 |
|---|---|---|---|---|---|---|---|---|---|
| A | No anthropometric failure | Normal WAZ, HAZ, and WHZ | No | No | No | 38.62 | 43.42 | 48.42 | 53.51 |
| B | Wasting only | WHZ<-2SD but normal WAZ and HAZ | Yes | No | No | 1.14 | 2.36 | 2.66 | 3.69 |
| C | Wasting and underweight | WHZ and WAZ<-2SD but normal HAZ | Yes | No | yes | 4.08 | 4.07 | 3.22 | 3.34 |
| D | Wasting, underweight, and stunting | WHZ, WAZ, and HAZ<-2SD | Yes | Yes | Yes | 5.50 | 4.10 | 3.97 | 3.08 |
| E | Stunting and underweight | HAZ and WAZ <-2SD but normal WHZ | No | Yes | Yes | 32.65 | 26.82 | 20.24 | 15.76 |
| F | Stunting only | HAZ<-2SD but normal WAZ and HWZ | No | Yes | No | 13.07 | 15.58 | 20.12 | 19.47 |
| Y | Underweight only | WAZ<-2SD but normal HAZ and WHZ | No | No | Yes | 4.87 | 3.50 | 1.37 | 1.16 |
| Stunting | D+E+F | Height-for-age (HAZ<-2SD) | | | | 51.22 | 46.50 | 44.30 | 38.30 |
| Wasting | B+C+D | weight-for-height (WHZ<-2SD) | | | | 10.70 | 10.50 | 9.90 | 10.10 |
| Underweight | C+D+E+Y | Weight-for-age (WAZ<-2SD) | | | | 47.10 | 38.50 | 28.80 | 23.30 |
| CIAF | B+C+D+E+F+Y (1-A) | Composite Index of Anthropometric Failure (CAIF) | | | | 61.38 | 56.58 | 51.58 | 46.49 |

the crude and BLUP estimates of Zones were merged with the shapefiles of the country to display the performances of each Zone using maps.

## Results

The crude prevalence of different measures of undernutrition status in children aged 0–59 months for the years 2000, 2005, 2011, and 2016 in Ethiopia was summarized in Table 1 and Fig 2. In the EDHS of 2000 61.38% of the children have one or more forms of under-nutrition (CIAF) with stunting (51.22%), and underweight (47.10%); stunting only (13.07%). From time to time, except for stunting only, the prevalence of all anthropometric measures was declined (Table 1). Specifically, the prevalence of the conventional under-nutrition measures and CIAF is decreased from time to time. Moreover, the CIAF is higher than the other measures, which indicated the real burden of the child's under-nutrition status in the country Fig 2.

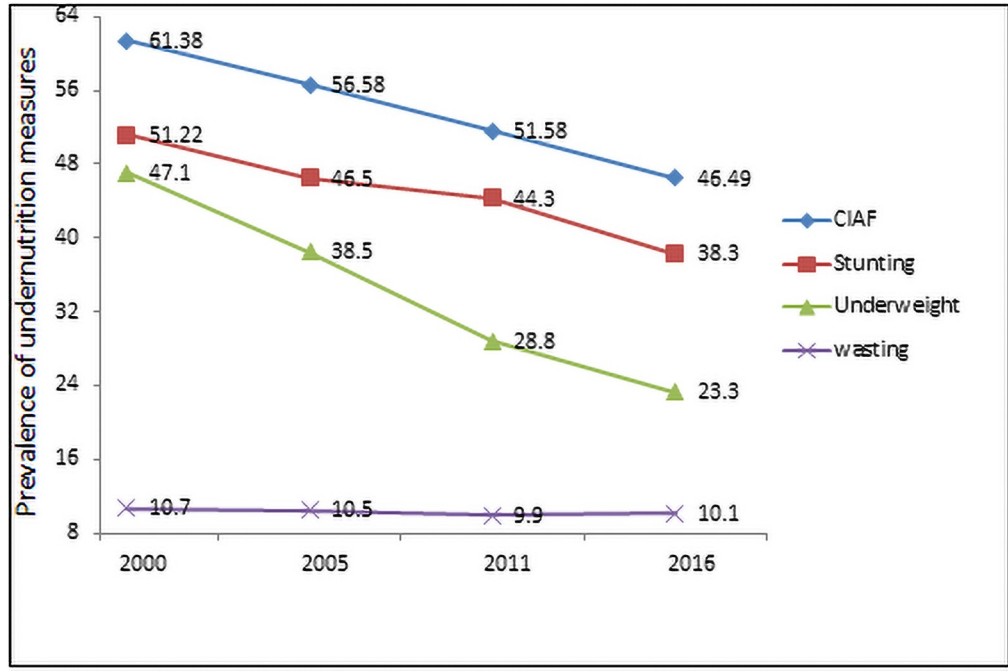

**Fig 2. Distribution of under-nutrition prevalence status among U5C in Ethiopia in four consecutive survey years.**

**Table 2. The CIAF prevalence and their 95% CI for each of the child characteristics in Ethiopia, 2000–2016.**

| Variables | 2000 EDHS | 2005 EDHS | 2011 EDHS | 2016 EDHS | 2000–2016 |
|---|---|---|---|---|---|
| Outcome variables | | | | | pooled |
| Childhood (CIAF) | 61.3 (59.4, 63.1) | 56.7 (54.3, 59.0) | 51.9 (50.1, 53.7) | 46.6 (44.6, 48.6) | 53.7 (52.6, 54.8) |
| **Child level covariates** | | | | | |
| Sex of child | | | | | |
| Male | 31.3 (29.9, 32.8) | 29.4 (27.5, 31.3) | 27.7 (26.3, 29.1) | 25.1 (23.7, 26.5) | 28.2 (27.4, 29.0) |
| Female | 23.0 (28.7, 31.3) | 27.3 (25.5, 29.2) | 24.2 (22.8, 25.7) | 21.5 (20.1, 23.0) | 25.6 (24.8, 26.3) |
| age of a child (months) | | | | | |
| <6 | 1.5 (1.2, 1.9) | 1.3 (0.9, 1.9) | 2.6 (2.1, 3.1) | 3.2 (2.7, 3.8) | 2.2 (2.0, 2.5) |
| 6–23 | 19.7 (18.6, 20.9) | 17.1 (15.5, 18.8) | 13.3 (12.4, 14.3) | 13.2 (12.1, 14.3) | 15.7 (15.1, 16.3) |
| 24–59 | 40.1 (38.6, 41.6) | 38.3 (36.3, 40.4) | 36.1 (34.5, 37.6) | 30.2 (28.6, 31.8) | 35.8 (34.9, 36.7) |
| Vitamin A | | | | | |
| Yes | 36.7 (34.3, 39.1) | 28.4 (25.5, 31.5) | 24.5 (22.7, 26.3) | 19.1 (17.5, 20.8) | 27.8 (26.7, 29.1) |
| No | 24.0 (22.6, 29.8) | 28.3 (25.6, 31.2) | 27.5 (25.8, 29.2) | 27.5 (25.8, 29.3) | 25.8 (24.8, 26.9) |
| Birth order | | | | | |
| 1 | 9.8 (9.0, 18.3) | 7.7 (6.6, 8.9) | 9.3 (8.4, 10.3) | 8.0 (7.0, 9.1) | 8.8 (8.3, 9.3) |
| 2–3 | 18.3 (17.1, 19.6) | 16.2 (14.8, 17.8) | 16.0 (14.9, 17.2) | 13.6 (12.3, 15.0) | 16.1 (15.4, 16.8) |
| 4+ | 33.1 (3.73, 34.5) | 32.7 (30.5, 35.1) | 26.6 (25.0, 28.3) | 24.9 (23.3, 26.7) | 28.8 (27.9, 29.7) |
| Breastfeeding | | | | | |
| Yes | 14.8 (13.6, 16.1) | 12.1 (10.8, 13.6) | 13.9 (12.7, 15.1) | 13.39 (12.11, 14.79) | 13.7 (13.1, 14.4) |
| No | 46.5 (44.6, 48.4) | 44.5 (42.3, 46.8) | 38.1 (36.3, 39.9) | 33.29 (31.4, 35.0) | 40.0 (38.9, 41.0) |
| Comorbidity | | | | | |
| Yes | 30.0 (28.4, 31.8) | 18.5 (16.8, 20.3) | 14.9 (13.7, 16.2) | 12.37 (11.2, 13.7) | 19.1 (18.1, 20.0) |
| No | 31.2 (29.7, 32.8) | 38.2 (35.5, 40.9) | 37.0 (35.2, 38.9) | 34.2 (32.4, 36.2) | 34.6 (33.7, 35.7) |
| Size of the child at birth | | | | | |
| Smaller than average | 22.2 (20.8, 23.7) | 18.1 (16.3, 20.0) | 17.6 (16.4, 19.0) | 14.5 (13.1, 16.0) | 18.1 (17.4, 18.9) |
| Average | 21.8 (20.6, 23.1) | 22.3 (20.6, 24.1) | 19.31 (18.1, 15.0) | 18.7 (17.3, 20.2) | 20.3 (19.6, 21.1) |
| Larger than average | 17.2 (15.9, 18.6) | 16.3 (14.7, 18.1) | 15.0 (13.5, 16.6) | 13.4 (12.14, 14.8) | 15.3 (14.5, 16.0) |
| Dietary diversity score (DDS) | | | | | |
| Below minimum | 59.6 (57.8, 61.4) | 51.3 (49.0, 53.6) | 29.1 (27.4, 30.8) | 26.4 (24.62, 28.3) | 40.4 (19.1, 41.8) |
| Minimum | 1.7 (1.3, 2.2) | 5.4 (4.45, 6.5) | 22.8 (21.2, 24.5) | 20.2 (18.69, 21.8) | 13.3 (12.4, 14.2) |
| Types of birth | | | | | |
| Singleton | 60.4 (58.5, 62.2) | 55.9 (53.5, 58.2) | 50.6 (48.7, 52.5) | 45.7 (42.9, 47.2) | 52.3 (51.3, 53.6) |
| Multiple | 0.9 (0.6, 1.7) | 0.8 (0.5, 1.4) | 1.4 (1.4, 2.4) | 1.5 (1.1, 2.1) | 1.3 (1.1, 1.5) |

The proportion (the 95% CI) of children having CIAF by the child and maternal-household related covariates for each of the survey years as well as for all the years combined is presented in Tables 2 and 3 respectively. The overall proportion of CIAF in Ethiopia was 53.7% [95% CI: 52.6, 54.8], with the highest proportion in 2000 EDHSs (61.3%) and the lowest proportion in 2016 (46.6%). In all EDHS datasets, the CIAF proportion of males, the higher age in months, the higher birth order, with comorbidity and having less than minimum dietary diversity children is higher than their counterparts (Table 2).

## The generalized linear mixed model

Table 4 represents the generalized linear mixed model with logit and probit link functions. Compared to the probit mixed mode, the model with logit link function fitted the data relatively well (lower AIC and BIC) [61]. Besides, the cross-zone variances for the logit model

**Table 3. Prevalence with 95% CI of CIAF with respect to household characteristics in Ethiopia for EDHS (2000, 2005, 2011, and 2016).**

| Variables | 2000 EDHS | 2005 EDHS | 2011 EDHS | 2016 EDHS | pooled (2000–2016) |
|---|---|---|---|---|---|
| Mother's age | | | | | |
| 15–24 | 14.7 (13.5, 15.9) | 12.5 (11.0, 14.2) | 11.9 (10.7, 13.2) | 10.4 (9.2, 11.7) | 12.3 (11.7, 13.1) |
| 25–34 | 29.7 (28.2, 31.2) | 28.0 (26.1, 30.0) | 27.2 (25.7, 28.8) | 24.1 (22.5, 25.8) | 27.1 (26.3, 28.0) |
| 35–49 | 17.1 (15.7, 18.2) | 16.2 (14.4, 18.2) | 12.8 (11.6, 14.0) | 12.1 (11.0, 13.3) | 14.2 (13.6, 14.9) |
| Place of residence | | | | | |
| Rural | 56.1 (53.8, 58.1) | 53.8 (51.4, 56.1) | 48.1 (46.1, 50.1) | 42.9 (40.8, 45.0) | 49.5 (48.3, 50.7) |
| Urban | 5.3 (4.0, 7.1) | 2.9 (2.1, 4.1) | 3.8 (3.0, 48.3) | 3.7 (2.9, 4.7) | 4.19 (3.5, 5.0) |
| Mother's education | | | | | |
| No formal education | 51.6 (49.5, 53.7) | 46.70(44.0, 49.4) | 37.9 (35.9, 40.0) | 32.8 (30.7, 34.9) | 41.6 (40.3, 42.8) |
| Primary | 7.4 (6.5, 8.5) | 8.7 (7.3, 10.2) | 13.3 (11.9, 14.9) | 12.0 (10.7, 13.5) | 10.6 (9.9, 11.3) |
| Secondary and above | 2.2 (1.7, 2.9) | 1.3 (0.9, 1.9) | 0.7 (0.47, 1.0) | 1.8 (1.5, 2.2) | 1.6 (1.3, 1.8) |
| Father's education | | | | | |
| No formal education | 41.6 (39.3, 43.8) | 34.8 (32.3, 37.4) | 28.1 (26.1, 30.2) | 23.5 (21.5, 25.6) | 31.5 (30.3, 32.8) |
| Primary | 14.3 (12.9, 15.8) | 17.0 (15.4, 18.8) | 21.1 (19.6, 22.7) | 16.8 (15.1, 18.6) | 17.4 (16.6, 18.3) |
| Secondary and above | 5.45 (4.6, 6.5) | 4.9 (3.9, 6.1) | 2.8 (2.2, 3.4) | 6.3 (5.4, 7.3) | 4.8 (4.4, 5.3) |
| Woman's autonomy | | | | | |
| Low autonomy | 36.9 (35.0, 38.9) | 23.4 (21.2, 25.6) | 22.0 (19.3, 22.70) | 18.8 (17.1, 20.6) | 25.4 (24.3, 26.5) |
| Middle autonomy | 11.1 (10.0, 12.2) | 23.1 (21.0, 25.1) | 18.2 (16.8, 19.6) | 17.0 (15.7, 18.5) | 16.4 (15.6, 17.2) |
| High autonomy | 13.3 (11.9, 14.7) | 10.4 (8.8, 12.2) | 12.8 (11.4, 14.2) | 10.8 (9.6, 12.1) | 11.9 (11.2, 12.7) |
| Source of drinking water | | | | | |
| unimproved | 20.1 (17.5, 22.7) | 24.6 (22.1, 27.4) | 28.3 (25.5, 31.2) | 21.3 (18.5, 24.4) | 23.2 (21.7, 24.7) |
| improved | 41.3 (38.6, 44.1) | 32.1 (29.4, 34.9) | 23.6 (21.0, 26.5) | 25.3 (22.6, 28.2) | 30.5 (29.0, 32.1) |
| Toilet facilities | | | | | |
| unimproved | 53.9 (51.7, 56.1) | 40.7 (38.1, 43.5) | 27.5 (25.3, 30.0) | 20.3 (17.5, 23.5) | 35.1 (33.5, 36.8) |
| improved | 7.4 (5.9, 9.1) | 15.9 (13.9, 18.2) | 24.5 (22.3, 26.7) | 26.3 (23.7, 29.1) | 18.6 (17.4, 19.9) |
| BMI | | | | | |
| Underweight | 15.2 (13.9, 16.5) | 12.7 (1.26, 14.3) | 12.4 (11.2, 13.7) | 10.3 (9.2, 11.4) | 12.5 (11.9, 13.2) |
| Normal | 45.0 (43.3, 46.8) | 42.4 (39.9, 44.9) | 38.2 (36.5, 39.9) | 34.6 (32.4, 36.8) | 40.0 (38.7, 40.8) |
| Overweight | 1.0 (0.7, 1.5) | 1.6 (1.10, 2.4) | 1.3 (1.3, 1.7) | 1.8 (1.4, 2.2) | 1.4 (1.2, 1.6) |
| Household number | | | | | |
| Less than 4 | 14.3 (13.1, 15.6) | 11.3 (9.96, 12.7) | 11.94 (10.70, 13.30) | 11.3 (10.0, 12.8) | 12.3 (11.7, 13.0) |
| 5–9 | 42.2 (40.5, 43.9) | 41.7 (39.5, 44.0) | 36.74 (34.98, 38.55) | 32.4 (30.7, 34.2) | 37.7 (36.7, 38.7) |
| ≥10 | 4.8 (4.0, 5.7) | 3.9 (2.9, 4.7) | 3.24 (2.67, 3.93) | 2.9 (2.2, 3.7) | 3.7 (3.3, 4.1) |
| Number of U5C in HH | | | | | |
| 1 | 19.9 (18.6, 21.3) | 18.1 (16.4, 19.9) | 17.9 (16.6, 19.3) | 16.6 (15.0, 18.3) | 18.0 (17.2, 18.8) |
| 2 | 32.7 (31.0, 34.4) | 29.2 (27.0, 31.6) | 25.8 (24.1, 27.6) | 22.0 (20.4, 23.8) | 27.2 (26.2, 28.2) |
| 3 or more | 8.6 (7.6, 9.8) | 9.3 (7.92, 11.0) | 8.3 (7.2, 9.4) | 8.0 (6.8, 9.4) | 8.5 (7.9, 9.2) |
| Media exposure | | | | | |
| No | 47.8 (45.7, 49.0) | 37.8 (35.1, 40.6) | 22.8 (21.0, 24.7) | 33.3 (31.0, 35.7) | 35.2 (33.8, 26.6) |
| Yes | 13.5 (12.3, 14.8) | 18.9 (16.7, 21.2) | 29.1 (27.4, 31.0) | 13.3 (11.9, 14.9) | 18.5 (17.5, 19.6) |
| Mother's working status | | | | | |
| Unemployed | 25.8 (23.5, 28.2) | 41.8 (39.3, 44.3) | 34.0 (31.8, 36.3) | 33.9 (31.9, 35.9) | 32.4 (31.1, 33.6) |
| Employed | 35.5 (33.0, 38.1) | 14.9 (12.8, 17.3) | 17.9 (16.0, 20.0) | 12.7 (11.3, 14.2) | 21.3 (20.1, 22.6) |
| Wealth quintile | | , | | | |
| Poorest | 8.5 (7.0, 10.2) | 13.3 (11.5, 15.4) | 13.1 (11.4, 15.0) | 12.5 (10.2, 15.3) | 11.6 (10.6, 12.7) |
| Poorer | 17.0 (15.6, 18.6) | 13.8 (12.1, 15.8) | 12.7 (11.3, 14.1) | 11.9 (10.6, 13.4) | 13.8 (13.1, 14.6) |
| Middle | 12.3 (10.9, 13.4) | 12.1 (10.4, 13.8) | 11.3 (10.0, 12.7) | 10.0 (8.7, 11.4) | 11.3 (10.6, 12.1) |

*(Continued)*

**Table 3.** (*Continued*)

| Variables | 2000 EDHS | 2005 EDHS | 2011 EDHS | 2016 EDHS | pooled (2000–2016) |
|---|---|---|---|---|---|
| Richer | 11.3 (10.1, 12.7) | 11.1 (9.6, 12.7) | 10.2 (8.8, 11.8) | 7.6 (6.4, 8.9) | 9.8 (9.1, 10.6) |
| Richest | 12.2 (10.6, 14.0) | 6.5 (5.3, 8.0) | 4.7 (3.9, 5.7) | 4.6 (3.8, 5.5) | 7.2 (6.4, 7.9) |

Children from an older age of mothers, rural, uneducated parents, and low autonomy of mothers, unimproved toilet facilities, and lower wealth index had a higher proportion of CIAF compared to their counterparts (Table 3).

**Table 4. Comparison of models to assess the random effect (zones).**

| | Model statistic | | Random component variance parameter estimates | | |
|---|---|---|---|---|---|
| Models | AIC | BIC | $\sigma_u$ | ICC | p-value |
| *GLMM logit* | *37617.64* | *37949.96* | *0.196* | *0.056* | *<0.001* |
| GLMM probit | 37764.02 | 38096.34 | 0.119 | 0.035 | <0.001 |

($\sigma_u = 0.196$) was larger than the probit model ($\sigma_u = 0.119$). Moreover, the random effect is statistically significant and inclusion of the zone as a random effect in the model helped to explain substantially the variability unaccounted for in the single-level logistic regression models (SLMs). Hence, the GLMM was appropriate for analyzing CIAF among U5C in Ethiopia, and the mixed-effect logit model was preferred over the probit mixed model (Table 4).

Among the child-related covariates: age of a child in months, higher birth orders, and having extra diseases, had a statistically significant association with increased odds of CIAF among the under-five children. While female children and a larger size at birth were significantly associated with a decreased odds of CIAF among the under-five children. Specifically, the odds of CIAF among female children was 0.862 (95% CI: 0.813, 0.911) times lower compared to their male counterparts. Children aged 9–23 months, and 24–59 months had 1.689 (95% CI: 1.603, 1.755) and 1.889 (95% CI: 1.803, 1.971) times higher odds of CIAF compared to the reference group (<6 months). The odds of CIAF among multiple births were 2.032 (95% CI: 1.841, 2.223) times higher than those of singleton births (Fig 3).

Among the maternal-household related covariates: children from rural areas, higher maternal age, lower educational levels of parents, lower autonomy of mothers, employed mothers, from unimproved toilet facilities, and oldest survey years are generally associated with lower odds of CIAF among the under-five children. Particularly, the odds of CIAF among children living in the rural residence were 1.211 (95% CI: 1.118, 1.304) times as compared to their urban counterparts. A child born to mothers who had primary, and secondary or above education level had 0.922 (95% CI0.853, 0.990) and 0.589 (95% CI: 0.459, 0.720) times less likely to have CIAF than a child born to mothers who had no formal education respectively. Moreover, children born from educated fathers (primary, secondary, or above) had lower odds compared to those from fathers who had no formal education. Children born to mothers who have media access had a lower odds of being CIAF (0.932) as compared to children born from mothers who had no media access. Children born from the middle, rich, and richest wealth quintile households had 0.885 (95% CI: 0.803, 0.967), 0.820 (95% CI: 0.737,0.902), and 0.726 (95% CI: 0.632, 0.819) times less odds of CIAF compared to those born to the poorest household wealth index respectively. Compared with 2000 EDHS, children in the years 2005, 2011, and 2016 205 were associated with a lower prevalence of CIAF by 12.4%, 23%, and 29.7% respectively (Fig 4).

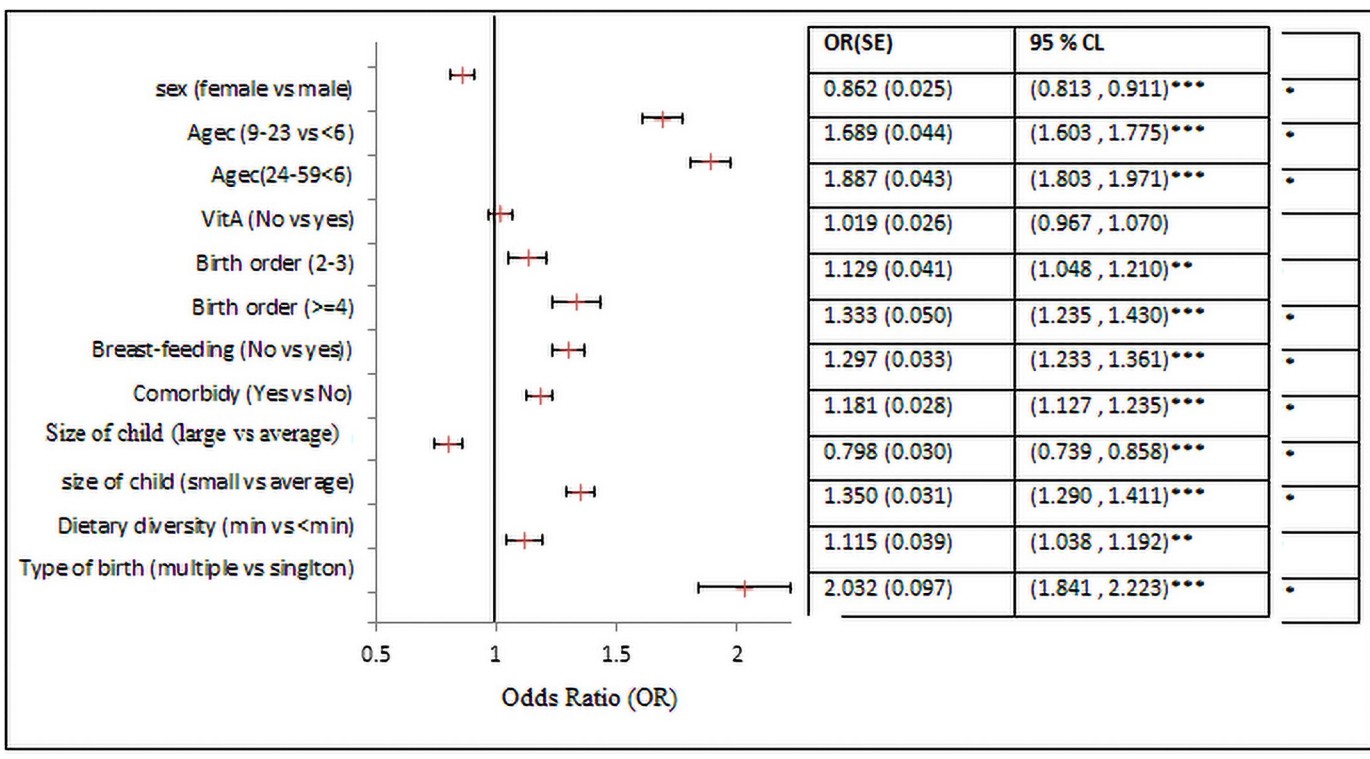

**Fig 3. The fixed effects of childhood covariates.**

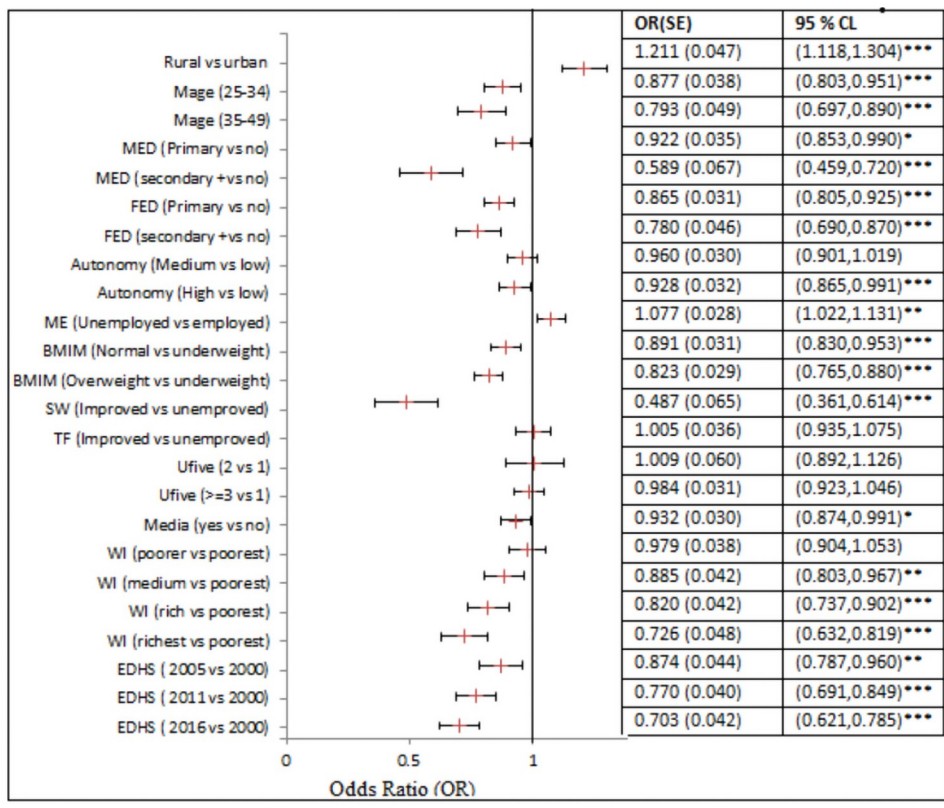

**Fig 4. The fixed effects covariates related to maternal and household characteristics.**

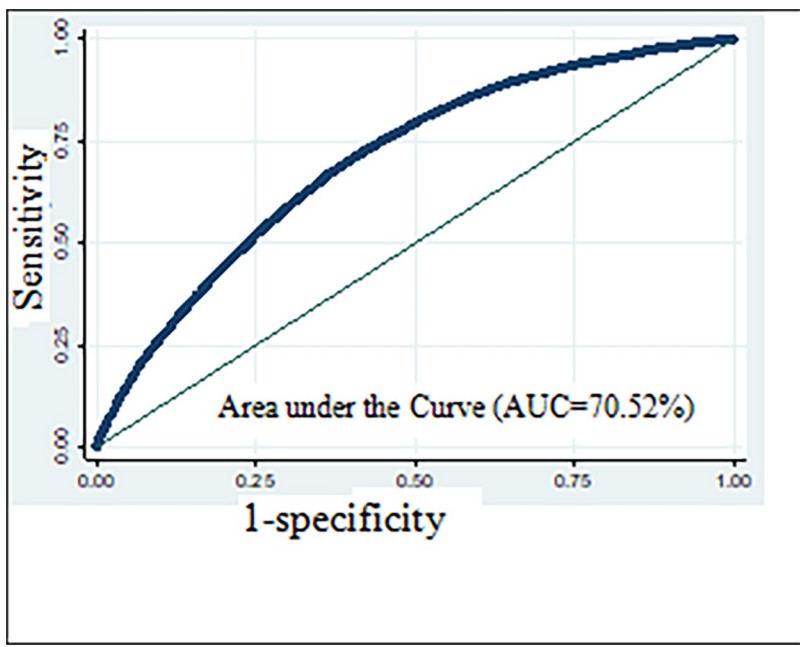

**Fig 5. Area under the receiver operating characteristic (AUC) curve for CIAF classification.**

## Model diagnostics

Model diagnostics were done for the final model using various model diagnostics techniques. Though this study considered the potential interaction effects of several covariates on CIAF, no interaction effects were found significant. The predicting probability of the model (model accuracy) was evaluated using Receiver Operating Characteristic (ROC) ("Fig 5") which was 70.52%, indicating that the model was good enough in differentiating children having CIAF from those not having it correctly [62] (Fig 5).

## Crude prevalence and BLUP of CIAF

The crude prevalence and the estimated BLUP of CIAF for under-five children have been pictorially presented across 72 Zones of the country. The empirical kriging output was mapped to the crude prevalence of undernutrition measures interpolating the available data to the areas where data were not taken. The red and blue colors indicated the areas with the highest and lowest crude prevalence of CIAF respectively (Fig 6). The maps show that there were wide zonal disparities in undernutrition crude prevalence (CIAF) in Ethiopia. The lowest prevalence was observed in the central part of the country including Addis Ababa, Somali, and Dire-Dawa, while the highest was observed in the northern part and some parts of southern Ethiopia.

One of the objectives of this study was to compare the performance of zones on the CIAF among U5C in Ethiopia using GLMM. The blue color is the best performing Zone and the red is the worst performing Zone in terms of undernutrition (CIAF) improvement. The negative BLUP is associated with decreased odds of CIAF in the Zones, while a positive BLUP is associated with increased odds of CIAF in the Zones [54, 63] (Fig 6).

Based on the standardized BLUP estimates, the zones were ranked and the best five (those with lowest standardized BLUP values) and top 5 "worst" (those with the highest standardized

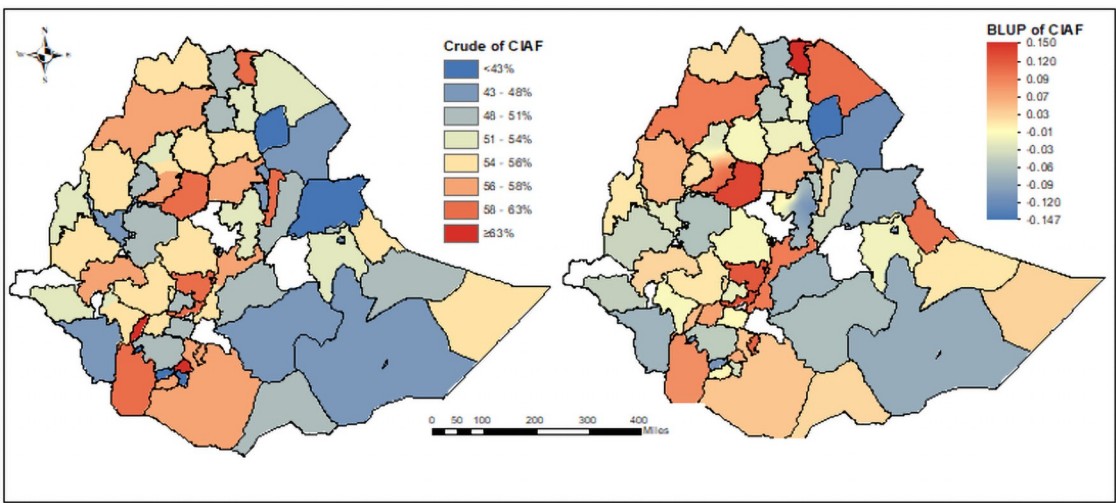

**Fig 6. The crude and estimated CIAF among U5C in Ethiopian administrative zones.**

BLUP values) performing zones in terms of CIAF respectively were selected. Accordingly, Welwel in Somali; West Wellega and West Shewa in Oromia, all zones in Addis Ababa and Zone 2 in Afar region were the top "best"; Gedeo and Yem Special Woreda in SNNP; south Gondar, East and West Gojjam in Amhara region were the top "Worst" performing zones in terms of CIAF among U5C respectively in Ethiopia. Both the crude prevalence and BLUP showed a high degree of variation in CIAF at zonal levels in Ethiopia.

## Discussion

This study provides new estimates for the prevalence of undernutrition by aggregating traditional undernutrition indices and applying them to an Ethiopian population dataset. Our findings show that when conventional indices (stunting, wasting, and underweight) are used alone, they miss a significant number of under-five children who already have multiple anthropometric deficits.

This issue was avoided by employing the CIAF aggregate measures of malnutrition [5]. In this study, the CIAF was utilized for the first time to provide an overall estimate of the undernutrition of U5C in Ethiopia over the last two decades. Studies conducted in Ethiopia [64–66] focused on either of the conventional indices may be suitable to inform interventions targeting at the reduction of each of the conventional indices alone, whereby, this study has the potential to ameliorate main drivers of all forms of undernutrition in the country. A generalized mixed model was adopted to assess the prevalence of CIAF and its associated risk factors among U5C in Ethiopia over time by accounting for the spatial heterogeneity of childhood CIAF at the zonal level. The prevalence of CIAF was higher across all EDHS survey years when compared to the prevalence of conventional measures [67, 68]. Even though the prevalence of CIAF in Ethiopia has decreased from 61.38% to 46.49% over time, it remains higher than countries such as India [69], China [70], Bangladesh [10], Tanzania [11], Malawi [71], Myanmar [72] and other SSA countries [73–76], and it is lower than studies conducted in Yemen [4] and southern India [77].

Ethiopia is one of the countries with the highest prevalence of undernutrition in SSA countries [64–66], which indicates that childhood undernutrition in the country needs urgent

attention. Despite progress, the situation of CIAF among under-five children in Ethiopia remains a major public health concern.

The CIAF was found to be strongly related to child, maternal, and household-related factors in this study. The findings indicated that compared to male children from a similar socioeconomic background, female children were less likely to have CIAF. This study is consistent with previous studies [26, 78, 79] and meta-analysis on gender and undernutrition indicated that boys in SSA were more likely to be undernourished than girls in early childhood stages [8, 78], which might be due to the biological growth and vulnerability of males morbidity in early infancy [80]. Besides, there is a perception that girls are less likely to be influenced by environmental stress than boys [26, 78].

The result revealed that children living in rural areas have more prevalence of CIAF compared to urban children. This is consistent with previous studies conducted in Myanmar [72], Bangladesh [10], this could be because children in cities have better living conditions and easier access to food. Furthermore, our findings show that children from lower-income households are more likely to be affected by the CIAF than their counterparts from higher-income households. This is inlined with the study conducted previously in different countries [10, 11, 71]. This might be because the wealthiest households afford to purchase various qualities and quantities of food to feed their children and the access to health care services may be limited in poorer households compared to the richer ones.

The study also revealed that children in the older age group had a higher risk of CIAF than the Youngers age group. This is consistent with studies conducted in different countries like Tanzania and Yemen [4, 11]. This could be due to a child receiving a more nutritious and balanced diet at a younger age, but as a child grows older, the discontinuation of breastfeeding and an increase in nutritional demand could be possible reasons. Furthermore, the results showed that children born to thin mothers had higher CIAF than children born to mothers with normal BMI, which is consistent with other studies conducted in SSA [11, 71] and others [10, 11, 69, 71]. The poor nutrition status of mothers leads to children's low birth weight and hence causes of CIAF. Moreover, the higher the educational level of both mothers and fathers, the lower the risk of their children's undernutrition status. This could be because educated parents are more likely to follow basic nutrition and hygiene practices, which may reduce anthropometric failure. Another possible reason is that educated parents understand nutrition information provided by the media or health care providers, which could be a factor in preventing undernutrition [4, 11, 71].

The study showed significant spatial heterogeneity of childhood CIAF within and between the Zones. This difference may be due to climatic variation in geographical, political, and socio-cultural norms, and due to dietary-related factors in different Zones of Ethiopia [9, 79, 81].

Based on the BLUP estimates, this study incorporated Zones of residence as a random effect which allowed ranking the performance of the Zones on the odds of CIAF.

Most of the worst-performing zones in Ethiopia were found in the northern parts of the country. This could be due to the knowledge, attitude, and practices (KAP) gap on the part of the households on how to feed their children and themselves. In most regions of the country, it is a common practice to sell more food items such as legumes (beans, peas, and chickpeas), sheep, goat, cattle, milk, and milk products since they earn better income by selling these agricultural products rather than feeding these nutritious food items to their children [82, 83]. In most parts of the northern regions of the country, food insecurity and less number of meals the child eats per day(meal frequency) have detrimental effects in protecting child undernutrition. Along with the low level of knowledge, attitude, and practices about eating diverse and nutritious food, a significant proportion of populations in the northern region were drought-

prone. Waghimra zone, some districts of north and south Gondar, many districts of north and south Wollo are highly degraded areas and experience a frequent wave of drought. In the southern part of Ethiopia, the Gedeo zone was the most affected area in CIAF. This could be due to the continuous displacement and internal conflicts of the communities [27, 32, 84, 85].

Best performing areas were found in the capital Addis Ababa, and some parts of the Oromia region including (West Shewa and Wellega). This could be due to urbanization, level of maternal education, empowerment of women, and improvement in the KAP, and access to information from mass media that aware households to feed their children properly [27, 32, 82–85]. The culture of society contributes to the dietary practice besides the aforementioned factors.

Hence, the ranking of the BLUP allows the worst-performing Zones to be targeted to improve their undernutrition control strategies and allow the best performing Zones to be identified as good practices for implementation.

## Conclusion

Most prior studies focused on assessing the undernutrition status using stunting, underweight, and wasting separately. This study indicates that focusing on any one of these indicators underestimates the overall prevalence of undernutrition, which can be better captured by using CIAF. There is a decline in the overall burden of undernutrition in Ethiopia as measured by CIAF

Furthermore, the study found that children in the country with a higher wealth index, educated parents, higher birth weight, a mother with a normal BMI, and better sanitation are at a lower risk of having CIAF than their counterparts. Moreover, this study demonstrated that zone-specific variation is critical in modeling the risk of CIAF. In addition, unlike previous studies that focused on the first administrative areas (regions), the current study focused on the second administrative zonal level, where the majority of political decisions on various issues were made.

The strength of this work is that it used BLUP to rank each Zone's performance for the CIAF of children under the age of five over time. As a result, the Zones with the best and worst performance were identified. The study suggests further research into Zones that have not made progress in overcoming childhood undernutrition (CIAF). Moreover, the identified socio-demographic characteristics and zones at a higher risk of CIAF can be used to inform localized intervention and prevention strategies to improve children's nutritional status and health care in Ethiopia.

## Acknowledgments

The datasets used in this study were obtained from the DHS program thanks to the authorization received to download the dataset on the website.

## Author Contributions

**Conceptualization:** Haile Mekonnen Fenta, Temesgen Zewotir.

**Formal analysis:** Haile Mekonnen Fenta.

**Methodology:** Haile Mekonnen Fenta, Temesgen Zewotir, Essey Kebede Muluneh.

**Software:** Haile Mekonnen Fenta, Temesgen Zewotir, Essey Kebede Muluneh.

**Visualization:** Haile Mekonnen Fenta, Temesgen Zewotir, Essey Kebede Muluneh.

**Writing – original draft:** Haile Mekonnen Fenta, Essey Kebede Muluneh.

**Writing – review & editing:** Temesgen Zewotir.

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
