## [Decision Letter · Decision Letter 0]

25 Jun 2021

PONE-D-21-04022

Disparities in childhood Composite Index of Anthropometric Failure prevalence determinants across Ethiopian administrative Zones over time

PLOS ONE

Dear Dr. Fena,

Thank you for submitting your manuscript to PLOS ONE. After careful consideration, we feel that it has merit but does not fully meet PLOS ONE’s publication criteria as it currently stands. Therefore, we invite you to submit a revised version of the manuscript that addresses the points raised during the review process.

Please attend carefully to the queries raised by the reviewers regarding the methods used in your study and to the clarifications and presentational adjustments they have recommended. 

We look forward to receiving your revised manuscript.

Kind regards,

Jamie Males

Staff Editor

PLOS ONE

Journal Requirements:

1. Please ensure that your manuscript meets PLOS ONE's style requirements, including those for file naming. The PLOS ONE style templates can be found athttps://journals.plos.org/plosone/s/file?id=wjVg/PLOSOne_formatting_sample_main_body.pdf and https://journals.plos.org/plosone/s/file?id=ba62/PLOSOne_formatting_sample_title_authors_affiliations.pdf

2. Thank you for providing the date(s) when patient medical information was initially recorded. Please also include the date(s) on which your research team accessed the databases/records to obtain the retrospective data used in your study.

Additional Editor Comments (if provided):

Reviewers' comments:

Reviewer's Responses to Questions

**Comments to the Author**

1. Is the manuscript technically sound, and do the data support the conclusions?

Reviewer #1: Yes

Reviewer #2: Yes

2. Has the statistical analysis been performed appropriately and rigorously? 

Reviewer #1: Yes

Reviewer #2: Yes

3. Have the authors made all data underlying the findings in their manuscript fully available?

Reviewer #1: No

Reviewer #2: No

4. Is the manuscript presented in an intelligible fashion and written in standard English?

Reviewer #1: Yes

Reviewer #2: No

5. Review Comments to the Author

Reviewer #1: Overall, the manuscript is interesting. However, the manuscript might benefit from some revision.

The authors need to show a clear constructing of the composite measure index and compare the composite measure index and the three traditional measures indices (WAZ, HAZ, WHZ). Also, authors should address the question, "what does the difference between the composite indicator measure and the traditional once? In addition, as they using hierarchical data structure and generalized multilevel mixed model is selected (line 182), the authors need to provide a clear level of the model and the notation of the variables in each level in the method section.

DHS children's data contains children of women born in the last 5 years, hence, authors how the correlated effects of children from the same women treated? It needs clear explanations. On top of these, the title is “Disparities in childhood Composite Index of Anthropometric Failure prevalence determinants across Ethiopian administrative Zones over time” but the effect of time is not investigated/showed and hence authors should indicated the effect of time on the CIAF prevalence in their results.

Minor: all figures and tables are not clear and needs re-organize with PLOS ONE format. And the references are not following the journal format (e.g.: it is found in font size of 11 and single spacing). Abbreviations with its long form should state at the first time and used only abbreviations after a while, for instance “Composite Index of Anthropometric Failure (CIAF)” is used in whole document with similar fashion whereas “U5C” is in other round way.

Abstract:

1. Line 6-8: “…..in 2000, 2005, 2011, and 2016” should read “…. (EDHSs) of 2000, 2005, 2011 and 2016, a population-based cross-sectional study of 29,599 under-five year children from 63 Zones.”

2. Line 14: “The pooled prevalence of CIAF…..” what “Pooled” indicates here? CIAF is already indicates the aggregate value, thus authors should read as “the prevalence of CIAF….”

Introduction:

1. Line 39-41: “….different forms of malnutrition” should read “….different forms of undernutrition”. Because, CIAF is computed from wasting, stunting and underweight only and malnutrition is beyond these three measures.

2. Line 55-57: “As the administrative Zones are mainly ethnic-based, ….” Is unrelated and unclear statement and should remove from the section.

Methods:

1. Line 74-75: “This study was conducted on 29,599 children consisting……” does the study unit children?

2. Authors indicate somewhere predictors are considered from child level and women level and the linear complement of the model is given at line 110 wrongly. This is because, child level and women level predictors are not included in the model clearly. Hence it needs clarification!

3. Line 119-128: all of these lines are reading about BLUP, the model that authors used in this manuscript is GLMM and the parameter of such model is estimated by either MML or IRWLS or other but not by BLUP. Thus, authors are used wrong methods for parameter estimation and needs critical correction. In general, the model and parameter estimation method is not appropriate for the data and needs refitting and re-estimating the parameters using appropriate model (GMMM) and estimation method (MML or IGWLS or nay other).

Result:

1. It would have been useful for the authors to show what percentage or proportion children were classified as undernourished using each of the measures separately and what new information the composite measure is contributing.

2. Line 132: “ …. in the years between 2000 and 2016 in Ethiopia was summarized in Table 1” should read “….in the years 2000, 2005, 2011 and 2016 in Ethiopia was summarized in Table 1”, because between indicates that all years from 2000 to 2016.

3. Line 133-136: statements in these lines are vague and do not clearly stated which needs paraphrasing, for instance, what is (61.38, 46.49)% indicate?

4. Does it appropriate to have composite index for these three variables with the consideration of huge proportion difference among the outcome (stunting, wasting and underweight)? For example, there is huge difference on the proportion of stunting and underweight (see table 1). This should well explained in your result section.

Conclusion:

1. Be careful in the use of wording: for instance line 313 " Most prior studies focused on assessing the undernutrition status using stunting ….” Are you Shure that most are not considering the CIAF?

2. The key findings and implications should be summarized instead of repeating the results. Instead the conclusion needs to be made based on the findings and hence it needs correction and modification.

In addition, authors did not include all of the declaration sections and better to include all.

Reviewer #2: Disparities in childhood Composite Index of Anthropometric Failure prevalence

determinants across Ethiopian administrative Zones over time

Title

• The way the tile written is confusing. Is it disparities in prevalence or determinants? The way the finding was written is also not suggestive.

• Omit the third affiliation statement on the title page. i.e. “3Department of Statistics, College of Science, Bahir Dar University, Bahir Dar- Ethiopia”.

Abstract

• Your first statement on the background and the objective is relatively not conforming. Please make the background more specific.

• Under result part, I don’t think the word ‘femaleness’ as proper scientific term here. Better to report as ‘being female’ and also some other expressions like “no experience of comorbidity” is also not standard way of expressions.

• On the topic disparity was the main objective. However on the result it was explained as some supplementary finding.

• Your conclusion should be implication of your finding. For instance, the second statement under the conclusion cannot be you conclusion from this study by any means. Even it cannot be your finding. Rather it might be one justification to do this analysis.

• Revise your keywords. For instance “Adjusted odds ratio” may not be the keyword for this study

Background

• You haven’t explained anything about the determinants of anthropometric failure and why it is needed to study about the issue as it is one of the objective of this study.

• What do other studies say about the CIAF? It could be from study in the other countries.

• You said that administrative zones are ethnic based which is not the case in most parts of the country. In addition the idea of the whole sentence here is not coherent to what is written before it and difficult to understand its implication here with respect to your study

Method

• Line 63 remove subtitle “data sources”

• You have did wealth index for urban and rural using different assets for EDHS 2000. How did you combined after you constructed it for use as a variable?

• How did you scored women’s autonomy? Explain it briefly

• Was the data weighted? If any, how?

• How does CIAF pooled?

Result

• On line 141 you wrote table 2 and 3 as they are is reporting a two by two table results but table 2 is only describing one variable each through the EDHS years.

• Put percentages with one decimal place

• The way you put Childhood undernutrition (CIAF) on table 2 is misleading. For example 29,599 looks like total children with CIAF which is not the case.

• What is the justification behind classifying child age in the way you classified on table 2?

• Line 149- 152 “Compared to their counterparts, children in the lower age group, in the lower birth order, and without comorbidity associated with a lower prevalence of CIAF. However, the rate was higher among children with a smaller size at birth, and multiple birth types (Table 2)”. To make such findings meaningful you should put at least the confidence interval. Otherwise it is difficult rely on this simple figure to see the difference. For instance is 22.2 and 21.8 really statistically different for the birth order? Even 22.2 might not be different from 17.2 unless we see the confidence interval.

• Line 155, the expression “Less than half (45.83%) of the women had a lower level of autonomy of decision making in a range of health care situations.” is better if expressed in the other way i.e. “little above half (54.175) of women have average or above autonomy”

• The way you interpret ORs in the final model findings was not easily understandable and to the standard. Particularly, here you need to extensively edit for language.

• On model diagnostics, I am not clear with how you have used VIF to assess the multi-collinearity as this is particularly binary data.

Discussion

• Line 253-254 “This difference may be due to ethnic-based variation in geographical, political, and socio-cultural norms…” I don’t think this explanation is evidence based. How many of those included zones were classified based on ethnic based?

• Line 289-290, “ In the Amhara region, it is a common practice to sell more nutritious food items. I think this is not special to Amhara region and may not be strong explanation for higher CIAF

References

• Check for reference #1 improper use of author initials and #6, #7; using institutions/affiliations just like the way we put the authors name

Figures

• Some of the texts on your figures are not visible

6. PLOS authors have the option to publish the peer review history of their article (what does this mean?). If published, this will include your full peer review and any attached files.

Reviewer #1: **Yes: **Demeke Lakew Workie

Reviewer #2: **Yes: **Mohammed Feyisso Shaka

---

## [Author Response · Author response to Decision Letter 0]

1 Jul 2021

We have incorporated all the comments and suggestions given by the reviewers.

---

## [Decision Letter · Decision Letter 1]

2 Aug 2021

PONE-D-21-04022R1

Disparities in childhood Composite Index of Anthropometric Failure prevalence and determinants across Ethiopian administrative Zones

PLOS ONE

Dear Dr. Fena,

Thank you for submitting your manuscript to PLOS ONE. After careful consideration, we feel that it has merit but does not fully meet PLOS ONE’s publication criteria as it currently stands. Therefore, we invite you to submit a revised version of the manuscript that addresses the points raised during the review process.

ACADEMIC EDITOR: Reviewer 2 still points to some major concerns, thus I am sending back this paper to you for a major revision.

We look forward to receiving your revised manuscript.

Kind regards,

Srinivas Goli, Ph.D.

Academic Editor

PLOS ONE

Additional Editor Comments (if provided):

Reviewer 2 still points to some major concerns, thus I am sending back this paper to you for a major revision.

Reviewers' comments:

Reviewer's Responses to Questions

**Comments to the Author**

1. If the authors have adequately addressed your comments raised in a previous round of review and you feel that this manuscript is now acceptable for publication, you may indicate that here to bypass the “Comments to the Author” section, enter your conflict of interest statement in the “Confidential to Editor” section, and submit your "Accept" recommendation.

Reviewer #1: All comments have been addressed

Reviewer #2: (No Response)

2. Is the manuscript technically sound, and do the data support the conclusions?

Reviewer #1: (No Response)

Reviewer #2: Yes

3. Has the statistical analysis been performed appropriately and rigorously? 

Reviewer #1: (No Response)

Reviewer #2: Yes

4. Have the authors made all data underlying the findings in their manuscript fully available?

Reviewer #1: (No Response)

Reviewer #2: (No Response)

5. Is the manuscript presented in an intelligible fashion and written in standard English?

Reviewer #1: (No Response)

Reviewer #2: No

6. Review Comments to the Author

Reviewer #1: I found that the authors have adequately addressed my comments raised. However, to increase the quality of the manuscript, I recommend a language editing by a native English speaker or by professional.

Reviewer #2: Disparities in childhood Composite Index of Anthropometric Failure prevalence

determinants across Ethiopian administrative Zones over time

Reviewer comment

For Editor

• There is some distortion on the pdf built. Some letters are missing from the sentences throughout the document.

For author

General

Language needs to be revised

Specific comments

Title

• It says “College of Science” for the first author. Is there such college in the University or mistaken?

Abstract

• Please re-read the corrections you made for the context …. “Being female of the child….” Also change “no comorbidity” to absence of comorbidity

• Please present the key findings for disparities

• Your conclusion should be implication of your finding. For instance, the second statement under the conclusion cannot be you conclusion from this study by any means. Even it cannot be your finding. Rather it might be one justification to do this analysis. “Ranking of the performance of the Zones help to target the worst performing, Zones for immediate intervention and the best performing Zones as a role model through implementing the best practice in the national strategy to achieve the SDG2030”. How could we reach to this conclusion the findings you have presented here? Even you removed the better conclusion that we can get from your finding “Disparities of CIAF were observed between and within the Ethiopian administrative Zones over time.” but what do this disparity imply to based on what you have found from the disparity and factors?

Background

• You haven’t explained anything about the determinants of anthropometric failure and why it is needed to study about the issue as it is one of the objective of this study. Which was not addressed on the previous revision.

Discussion

• Your discussion is not as strong as the specific findings of your study. You haven’t well interpreted your findings and the evidence synthesis from your findings was not well established. You simply put so many literatures without strongly discussing about the implication of the findings together. The way you put the explanation for some interpreted findings are merely a simple guess rather than trying to put solid suggestions

References

• Your reference also still needs revision. For instance the author name for #1, the way you used et al. please revise based on the requirement for the style you have used

7. PLOS authors have the option to publish the peer review history of their article (what does this mean?). If published, this will include your full peer review and any attached files.

Reviewer #1: **Yes: **Demeke Lakew Workie

Reviewer #2: **Yes: **Mohammed Feyisso Shaka

---

## [Author Response · Author response to Decision Letter 1]

7 Aug 2021

Thank you for your time and effort for the improvement of our manuscript. The revised manuscript is submitted here.

---

## [Editor Report · Decision Letter 2]

16 Aug 2021

Disparities in childhood Composite Index of Anthropometric Failure prevalence and determinants across Ethiopian administrative Zones

PONE-D-21-04022R2

Dear Dr. Fena,

We’re pleased to inform you that your manuscript has been judged scientifically suitable for publication and will be formally accepted for publication once it meets all outstanding technical requirements.

Kind regards,

Srinivas Goli, Ph.D.

Academic Editor

PLOS ONE
---

## [Editor Report · Acceptance letter]

26 Aug 2021

PONE-D-21-04022R2 

Disparities in childhood Composite Index of Anthropometric Failure prevalence and determinants across Ethiopian administrative Zones 

Dear Dr. Fenta:

I'm pleased to inform you that your manuscript has been deemed suitable for publication in PLOS ONE. Congratulations! Your manuscript is now with our production department. 

Kind regards, 

on behalf of

Dr. Srinivas Goli 

Academic Editor

PLOS ONE